# AI-based coral species discrimination: A case study of the *Siderastrea* Atlantic Complex

**Marcos Soares Barbeitos** [1][*], **Flávio Alberto Pérez**[1], **Julián Olaya-Restrepo** [1][‡], **Ana Paula Martins Winter**[1][‡], **João Batista Florindo**[2], **Estevão Esmi Laureano**[2]

**1** Laboratório de Evolução dos Organismos Marinhos, Departamento de Zoologia, Universidade Federal do Paraná, Curitiba, Brazil, **2** Departamento de Matemática Aplicada, Instituto de Matemática, Estatística e Computação Científica (IMECC), Universidade Estadual de Campinas, Campinas, Brazil

☯ These authors contributed equally to this work.
‡ JOR and APMW also contributed equally to this work.
* msbarbeitos@gmail.com

**Data Availability Statement:** All data are fully available without restriction. The links to access the

## Abstract

Species delimitation in hard corals remains controversial even after 250+ years of taxonomy. Confusing taxonomy in Scleractinia is not the result of sloppy work: clear boundaries are hard to draw because most diagnostic characters are quantitative and subjected to considerable morphological plasticity. In this study, we argue that taxonomists may actually be able to visually discriminate among morphospecies, but fail to translate their visual perception into accurate species descriptions. In this article, we introduce automated quantification of morphological traits using computer vision (Completed Local Binary Patterns—CLBP) and test its efficiency on the problematic genus *Siderastrea*. An artificial neural network employing fuzzy logic (Θ-FAM), intrinsically formulated to deal with soft and subtle decision boundaries, was used to factor *a priori* species identification uncertainty into the supervised classification procedure. Machine learning statistics demonstrate that automated species identification using CLBP and Θ-FAM outperformed the combination of traditional morphometric characters and Θ-FAM, and was also superior to CLBP+LDA (Linear Discriminant Analysis). These results suggest that human discrimination ability can be emulated by the association of computer vision and artificial intelligence, a potentially valuable tool to overcome taxonomic impediment to end users working on hard corals.

## Introduction

The motivation for this article stems from our own personal frustration when trying to identify species belonging to the so called "*Siderastrea* Complex of the Atlantic" [1–3] based on recent taxonomic keys [4, 5]. The brief taxonomic review of the genus in the Supporting Information Text in S1 File is chock-full of adjectives and adverbs such as "finely", "rarely", "unusual", "poorly", "deeper", "uncommon", etc., with no baseline to gauge the applicability of these definitions. To be fair, this is certainly a feature of many taxonomic keys and the trouble that those who lack proper taxonomic training face going through them has been termed "taxonomic impediment to end users" [6]. In the case of hard corals, the majority of diagnostic characters

data are the following: https://morphobank.org/index.php/MyProjects/Project/form/project_id/5258 and https://github.com/leom-ufpr/CLBP_tEFAM.

**Funding:** MSB - (FBPN 1040-20151) - Fundação O Boticário de Proteção à Natureza's - (https://www.fundacaogrupoboticario.org.br/) JBF - (Grant #2020/01984-8) - São Paulo Research Foundation (FAPESP) - (https://fapesp.br/) JBF - (Grant #2020/09838-0) - São Paulo Research Foundation (FAPESP) - (https://fapesp.br/) JBF - (Grant #306030/2019-5) - National Council for Scientific and Technological Development, Brazil (CNPq) - (http://www.cnpq.br/) EEL - (Grant #2020/09838-0) - São Paulo Research Foundation (FAPESP) - (https://fapesp.br/) E.E.L. - (Grants #313313/2020-2) - National Council for Scientific and Technological Development - (http://www.cnpq.br/) ) The funders had no role in study design, data collection and analysis, decision to publish, or preparation of the manuscript.

**Competing interests:** The authors have declared that no competing interests exist.

are skeletal, continuous, highly variable within species and often correlated with environmental parameters [7]. Taxonomists do report ranges and/or averages and standard deviations of continuous measurements, such as chalice diameters. This should imprint some objectivity into the taxonomic work. However, these ranges frequently overlap among species and vary among biogeographic regions, adding even more uncertainty to the keys, lest in rare cases where specimens happen to fall well within the distributions of all diagnostic characters. We illustrate this problem in the genus *Siderastrea* using a simple exercise, as follows.

*Siderastrea* Blainville, 1830 (Scleractinia:Siderastreidae) is a genus that currently has four valid species: *Siderastrea savignyana* Milne Edwards & Haime, 1849 which occurs in the Western Pacific/Indian Ocean [8], and three Atlantic species, *Siderastrea radians* (Pallas, 1766), *Siderastrea siderea* (Ellis & Solander, 1786) and *Siderastrea stellata* Verril, 1868. The three latter species make up the Atlantic Complex. We compiled data from 3 morphometric studies of the genus [4, 9, 10] that reported averages and standard deviations (SD) for a number of continuous diagnostic characters. We computed confidence intervals (CI) for all of them as 1.96 x SD (i.e. assuming an underlying normal distribution). We graphed the results in S1 Fig in S1 File and the reasons behind our confusion are readily apparent. For instance, the distributions of distances among collumelae, number of septa and corallite diameters show more overlap between *Siderastrea siderea* from Panama and *Siderastrea stellata* from Paraíba state (Brazil) than what is found between *Siderastrea stellata* specimens collected from Paraíba and those collected from the neighboring state of Pernambuco. The character distributions for this last set of specimens are actually more similar to those sampled from *Siderastrea radians* collected in Panama (S1c, S1e, S1g Fig in S1 File). Collumella diameters, distances among corallites, theca thicknesses and columella depths all show biogeographic patterning, for they all seem significantly smaller in specimens collected in Mexico when compared to those sampled in the state of Bahia, Brazil (S1a, S1b, S1d and S1f Fig in S1 File). The reader will certainly find additional controversial patterns upon further examination of the figure.

In our personal experience, the fuzziness in coral species boundaries is in stark contrast with the clarity that specialists in this genus claim to have when following their own taxonomic keys. We argue that such vehemence is most likely not professional hubris: after looking at hundreds of coralla, taxonomists should be able to visually pick up subtle and hard-to-quantify differences in the arrangement of skeletal element. Their experience gives them an ability to discriminate among morphospecies that is much superior to those of an average end user. Technically speaking, they are probably picking up differences in skeletal texture. Image texture is the product of the spatial arrangement of image colors or intensities and is defined as the visual perception of the patterning or tactile sensation offered by a surface (e.g. smooth vs. rough). For instance, if a black-and-white image has 50% white pixels and 50% black pixels, the relative positioning of the pixels could render a rugged or leveled grayish surface, or generate a striped or checkerboard patterning [11]. Variation in patterning of coralla surfaces is dictated by the size and shape of many features related to corallite architecture and the integration of corallites within colonies, which are important in traditional taxonomy at the generic and specific levels [12] and hard to translate into natural language or conventional descriptive statistics, but may be adequately captured by texture descriptors (TD).

In this study, we attempted to emulate the taxonomists ability to discriminate among species in the *Siderastrea* Complex of the Atlantic by employing TD. Texture is an important visual attribute in automated image analysis that has been successfully applied, for instance, in the identification of plant species using leaf epidermis [13]. This is useful because it allows for the identification of plant species in the absence of reproductive structures, the main source of diagnostic characters in angiosperms, but not present year around. Unlike angiosperms, we have no clear-cut set of characters that we could employ in order to be sure of species

identification. Therefore, to factor our own uncertainty into the supervised classification procedure, we also employed Equivalent Measure Fuzzy Associative Memory (E-FAM) [14, 15]. Associative memory (AM) is a mathematical model designed to match input-output pairs called "fundamental memories". In our case, the input are TD and the output are the corresponding, tentatively identified species. This approach is inspired in the workings of the human brain: the image of an animal formed by our eyes is associated to the fundamental (or conceptual) memory we have of that animal (e.g. "cat", "horse", "dog", etc.). E-FAMs belong to a general class of associative memories called Θ-FAMs, fuzzy artificial neural networks (ANNs) with a competitive hidden layer [14]. This type of AM is useful when the association between image and concept is fuzzy, either because identification of an appropriate label (in our case, the species) is uncertain or because the descriptors are noisy. E-FAMs return the degree of membership, called **pertinence** of each object (image) to each possible class (species). Like probability, pertinence also ranges from 0 to 1, but it is otherwise unconstrained, so that an image may have identical pertinence to more than one species. The implementation of fuzzy logic into AM brings it even closer to the human perspective. For instance, a specialist may be unsure if the spotted cat photographed in captivity is a leopard or a jaguar because the image is out of focus. Coat patterning and cat shape may in fact be so blurry that she may not even entirely rule out the possibility that the feline is a cheetah, although she may consider it unlikely. This would be equivalent to stating that the image has high, identical pertinence to the "jaguar" *and* "leopard" classes, but only moderate pertinence to the "cheetah" class.

We employed robust machine learning (ML) performance statistics to benchmark the success of our chosen strategy against 3 different standards. The first was its application to a different genus (*Porites*) that has some species that can be easily distinguished on the basis of colonial shape and color (e.g. *P. astreoides* and *P. branneri*), but whose corallite architecture is extremely variable, displaying considerable overlap among different species (e.g. [16]) and also pronounced phenotypic plasticity (e.g. [17] and S2 Fig in S1 File). The purpose of this first experiment was to evaluate how the combination of CLBP+Θ-FAM fares using accurately labeled specimens (since labeling accuracy is of paramount importance to the success of image classification using IA [18]), but sampled from species whose corallite morphology is subjected to even greater intra and inter-specific variability than what is found in *Siderastrea* spp. In the second experiment, traditional morphometric descriptors sampled from *Siderastrea* spp. (some of which were used themselves in species identification), were also used as inputs to Θ-FAM. The purpose here was to evaluate if AI is powerful enough to discriminate among putative species using traditional characters or whether, as an unlearned user, performance would be hampered by limited and noisy data. Finally, we conducted supervised classification of TD using a variant of Linear Discriminant Analysis (LDA), a technique commonly used by taxonomists in dimension reduction and classification tasks. The purpose of this last experiment was to evaluate if fuzzy ANNs offers any advantage over a probabilistic approach familiar to many biologists.

## Materials and methods

### Sampling

Samples were obtained via free or SCUBA diving along 2500+ km of the Brazilian coast from Búzios (Rio de Janeiro State), Aracruz (Espírito Santo), Abrolhos Archipelago and Boipeba Island (Bahia), Tamandaré (Pernambuco) and Maxaranguape (Rio Grande do Norte). Sampling locations are mapped in Fig 1, further geographic information is provided in Table 1. We targeted 5 species: *Porites astreoides* Lamarck 1816, *Porites branneri* Rathbun 1888, *Siderastrea siderea* (Ellis & Solander, 1786), *Siderastrea stellata* Verril 1868 and *Siderastrea radians*

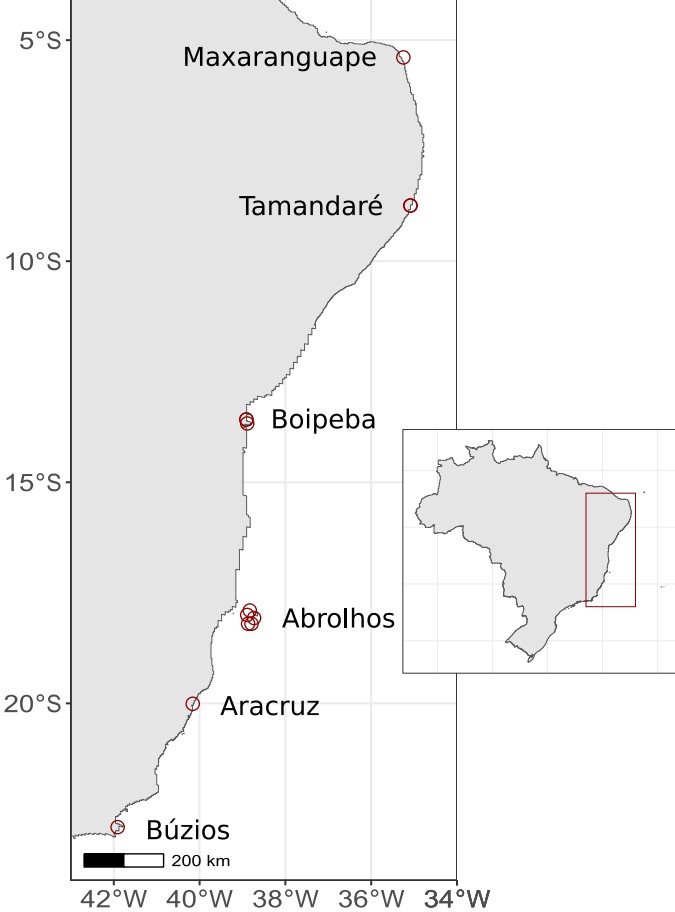

**Fig 1. Distribution of sampling locations along the Brazilian coast.** Further details in Table 1. We acknowledge the use of public domain shape files from IBGE/DGC—Base Cartográfica Contínua do Brasil, 1:250.000—BC250: v. 2017 (Rio de Janeiro, 2017).

(Palas 1766). Colony fragments were removed using hammer and chisel, placed in punctured, labeled plastic bags and left in salt water for a number of days until colonial tissue was mostly decomposed. Samples were then washed with tap water, placed in a new bag, and immersed in a solution of sodium hypochlorite in order to remove any remaining traces of tissue. After a second rinse, colonial skeletons were dried in an electric plant dryer for 15 hours before being deposited in the Collection of the Department of Zoology in the Federal University of Paraná. Collections were conducted under permits 50095–1 (Instituto Chico Mendes de Conservação da Biodiversidade—ICMBio), 004/2015 (Secretaria Municipal de Meio Ambiente e Pesca, Prefeitura da Cidade de Armação dos Búzios) and 004/2015 (Instituto de Desenvolvimento Sustentável e Meio Ambiente do Rio Grande do Norte—IDEMA).

## Species identification

*Porites* species can be easily distinguished in the field by colonial shape and color [19], although corallite structure is similar across species and varies across environments (see S2 Fig in S1 File). *Siderastrea* species were identified in the laboratory according to the most recent taxonomic treatment of the genus [4, 5], but identification was tentative given the lack of

**Table 1. Geographical distribution of sampled specimens.** CLBP = Complete Local Binary Pattern, Morpho = Traditional Morphometric Characters.

| State | Location | Site | S | W | Depth (m) | CLBP | | | | | Morpho. | | |
|---|---|---|---|---|---|---|---|---|---|---|---|---|---|
| | | | | | | *P. astreoides* | *P. branneri* | *S. cf. radians* | *S. cf. siderea* | *S. cf. stellata* | *S. cf. radians* | *S. cf. siderea* | *S. cf. stellata* |
| RJ | Búzios | Praia da Tartaruga | 22.80 | 41.91 | 0.5–2.0 | | | | | 10 | | | 17 |
| ES | Aracruz | Praia do Pichado | 20.01 | 40.16 | 0.5–1.5 | | | 2 | | 6 | 2 | | 13 |
| BA | Abrolhos | Caldeiras | 18.20 | 38.87 | 4.0–5.5 | 1 | | 3 | | 3 | 3 | | 2 |
| | | Chapeirinhos da Sueste | 18.20 | 38.79 | 8.0–12.0 | | | | 3 | 6 | | 3 | 5 |
| | | Chapeirões do Sul | 18.07 | 38.73 | 21.0–23.0 | | | | 4 | 1 | | 4 | 1 |
| | | Redonda | 18.00 | 38.89 | 5.0–6.0 | 1 | 3 | | | 2 | | | 1 |
| | | Chapeirão | 17.90 | 38.83 | 14.0–15.0 | 1 | | | | 2 | | | 3 |
| | Boipeba | Ponta dos Castelhanos | 13.67 | 38.89 | 0.5–2.0 | 1 | | | 1 | | | 1 | 4 |
| | | Praia de Moreré | 13.58 | 38.91 | 0.5–2.0 | 1 | | | | | | | |
| | | Tassimirim | 13.58 | 38.91 | 0.5–4.0 | 2 | 9 | 3 | | 3 | 3 | | 8 |
| PE | Tamandaré | Igreja de São Pedro | 8.74 | 35.09 | 0.5–2.0 | 2 | | | | | | | |
| | | Praia de Campas | 8.74 | 35.08 | 0.5–2.0 | 2 | | 5 | | | 6 | | |
| RN | Maxaranguape | Parrachos de Maracajaú | 5.39 | 35.25 | 0.5–2.0 | 4 | | 3 | 4 | 2 | 3 | 4 | 10 |
| Total | | | | | | 15 | 12 | 16 | 12 | 35 | 17 | 12 | 64 |

Columns S and W refer to latitude and longitude, in degrees and decimal degrees; Morpho. = Morphometrics. State acronyms: RJ—Rio de Janeiro, ES—Espírito Santo, BA—Bahia, PE—Pernambuco and RN—Rio Grande do Norte.

clear-cut diagnostic features. Colonies whose corallites were small, had on average less than 34 septa, stouter first cycle septa and absence of a complete fourth cycle were identified as *S. radians*. Colonies with larger corallites bearing on average more than 40 septa, slender first cycle septa, frequent occurrence of corallites with complete fourth cycles and no intra-tentacular budding were identified as *S. siderea*. The presence of intra-tentacular budding was considered a diagnostic character of *S. stellata*, although the absence of such character does not rule out the possibility that the colony belonged to this species. *S. stellata* colonies are very similar to *S. siderea*, although corallites in the first species are smaller and fourth cycle is complete in none but a few of them. S3 Fig in S1 File summarizes the breakdown of examined corallites according to the presence and completeness of septum cycles.

## Texture descriptors

Local Binary Pattern [20] (LBP) is a type of visual descriptor commonly used for classification in texture-oriented computer vision (i.e. pattern recognition by computer software). In gray-scale images, LBP generates "feature vectors" i.e., n-dimensional numerical vectors that represent some object, computed as relative differences between color-related quantities of a reference pixel and its neighbors. Completed Local Binary Patterns (CLBP) [21] is an extension of LBP that has the advantage of being insensitive to image orientation and luminosity, two parameters that are hard to control in scanning electronic microscopy. CLBP was the most efficient texture quantification strategy tested, as measured by the success in preliminary image classification using simple 5-fold cross-validation analysis employing K-nearest

neighbors with chi-square distances [22]. Mean classification success (±s.d.) for CLBP was 93.0±1.0% as opposed to 70.4±2.0% for fractal descriptors [23]. A formal description of this method as applied to this study was published elsewhere [24].

Texture quantification was applied to a subset of scanning electron micrographs (SEM) taken from 63 *Siderastrea* spp. and 27 *Porites* spp. colonies (Table 1). Micrographs were produced under a magnification of 11x (spanning more than one corallite, hence taken at the "colony scale") and a second set framing individual corallites, with variable magnification in the 20x range ("corallite scale"). Images were replicated 3 times per colony and scale (6 images per colony) totalling 532 micrographs (see S2 and S4 Figs S1 File for some examples).

## Traditional morphometrics

Morphometric measurements were carried out on micrographs taken under stereomicroscope of arbitrarily chosen corallites belonging to 93 *Siderastrea* spp. colonies (Table 1). Only mature corallites (5 per colony) were photographed, i.e. those with a complete third septal cycle, excluding those on the edges of the colonies. Micrographs were taken at the scale of individual corallites ($\sim$2.5X) and of their columellae ($\sim$6.3X). All measurements were obtained using ImageJ2 [25]. We quantified (1) corallite and (2) collumela diameters (calculated as the average of the largest and the smallest diameters), (3) mean thickness and (4) mean length of first cycle septa and (5) number of septa. We also scored, as binary characters, 4th septum cycle as (6) present/absent and, when present, (7) complete/incomplete, 5th cycle as (8) present/absent, (9) complete/incomplete and 6th cycle as (10) present/absent, (11) complete/incomplete. We took a second set of pictures from the same colonies aiming at including the largest possible number of corallites in the frame while keeping their columellar centers on the same planar projection. Due to the differences in corallite diameters and colonial curvature, the number of sampled corallites ranged from 6 to 28, averaging 17 across all colonies. XY coordinates of collumelar centers were also recorded using ImageJ2 and Delaunay triangulation [26] meshes connecting those centers were rendered using the R package DELDIR [27]. From those meshes, we computed the (12) minimum and (13) maximum distances between columellae for each colony. Colonies were also examined for the (14) presence/absence of intratentacular budding and (15) presence/absence continuity of septa between adjacent corallites. Morphometric characters are summarized in S1 Table in S1 File and continuous characters boxplots are graphed in S5 Fig in S1 File.

## Supervised classification

A formal description of Θ-FAM as applied to this study was published elsewhere [24]. We benchmarked the performance of the Θ-FAM applied to CLBP collected from *Siderastrea* spp. colonies against supervised Linear Discriminant Analysis (LDA) applied to the same descriptors. We chose LDA because this technique is commonly used by biologists in classification problems. It was not possible to apply LDA to traditional morphometric descriptors because many of the traits were not measured in a continuous scale, but binarily scored (i.e. presence/absence). The CLBP matrix is non-invertible because the number of descriptors is much larger than the number of samples in the matrix. It is also singular due to high correlation among some of the descriptors. These features prevent ordinary LDA from being applied to these data. Therefore, we employed a variant of LDA called DAPC (Discriminant Analysis of Principal Components) [28], in which a Principal Component Analysis (PCA) is first applied to the data. To prevent overfitting, a preliminary cross-validation procedure is used to choose the minimum number of components that maximize success in membership assignment of the validation set (up to a maximum of 10% of the data by default). Principal component selection

was performed using the *adegenet* package, implemented in R [29, 30] with *a priori* number of species (*K*) set to 3. Membership probabilities to each species were computed via the *lda* function implemented in the package *MASS* [31], also using 5-fold cross-validation.

## Classifier performance—TSS

Performances of Θ-FAM and DPCA applied to either CLBP or traditional descriptors were evaluated via 5-fold cross-validation experiments. In this procedure, images are randomly and equitably partitioned into 5 separate sets. Four sets are used to train the classifier and the fifth set is validated after training. The validated partition is swapped with another partition belonging to the training set, being subsequently validated. The procedure is repeated until all images are classified. To account for the stochastic error associated with image assignment, each experiment was repeated 100 times and classification success was quantified by computing True Skill Statistics [32] (TSS) for each replicate. TSS = E + S—1, where E and S are the replicate's specificity (True Positive Rate or TPR) and sensitivity (True Negative Rate or TNR), respectively. TSS ranges from -1 to 1, where 1 is the score of a perfect classifier, while classifiers no better than random have values equal to or lower than 0. Interpretation of TSS is similar to that of the Receiver Operating Characteristic (ROC) curve, generated by graphing TPR against False Positive Rates (FPR = 1—TNR) for a range of outcomes (i.e. the replicate's pertinence in the case of Θ-FAM or posterior probability in the case of DPCA). The area under the ROC curve (AUC) is normally used to express the overall efficiency of the classifier. It ranges from 0 (no better than random) to 1 (perfect classifier). AUCs have two major drawbacks: different curves obtained from classifiers with different performances may have identical areas and these areas may be also influenced by prevalence, i.e., the frequency of true positives in the data [32]. TSS is not subjected to such biases. Because there is no *a priori* cutoff value to determine if image classification was successful or not, TSS scores were computed for outcome values (pertinence scores or posterior probabilities) ranging from 0 to 1, in the same way that ROC curves are computed. Instead of AUCs, one may use the maximum TSS to express maximal classification success because it represents the best trade-off between sensitivity and specificity [32]. The classification score (pertinence or probability) that maximizes TSS in the training may be used as a threshold to binarize the classifier outcomes i.e., values equal to or greater than the threshold are scored as 1 when machine and human agree on image classification (true positives) and 0 otherwise. In order to prevent class imbalance when computing TSS scores, we randomly sampled scores for *n* images from each cross-validation experiment, where *n* corresponds to the number of images in the least sampled species in each genus, or the minority class in each tested combination of classifier and descriptor (i.e. CLBP+Θ-FAM—*Porites* spp., CLBP+Θ-FAM—*Siderastrea* spp., Morph+Θ-FAM—*Siderastrea* spp. and DPCA+CLBP—*Siderastrea* spp.). TSS scores were computed from sensitivities and specificities obtained using the function *roc* [33], implemented in R package *pROC*

It is important to highlight that TSS values were computed from multiclass confusion matrices, whose outputs differ from confusion matrices generated from binary classifiers. If an image of a colony identified as *S. cf. stellata* is labeled as *S. cf. stellata*, this outcome would be scored as a true positive for *S. cf. stellata* and as true negatives for the other two species. If it is however labeled as *S. cf. siderea*, the outcome would be scored as a false negative for *S. cf. stellata*, as a false positive for *S. cf. siderea*, but still as a true negative for *S. cf. radians*. Therefore, because outputs vary according to each class, it is only possible to estimate classification performance for each separate species and not for the classifier as a whole.

## Classifier performance—Meta-learning

When binarized outcomes of a skilled classifier are graphed against classification scores, the majority of true positives will be found above the binarization threshold. The relationship between binarized outcomes and classification scores may thus be described by a binary logistic regression that expresses the probability that the classifier will return a true positive for any given score. Logistic regressions may themselves be used as classifiers i.e., they may be trained to identify true positives using data of one or more predictive variables. They are called meta-classifiers when they (meta) "learn" from data generated by other classification algorithms [34]. Therefore, the predictive ability of binary logistic regressions fit to each of the four combinations of image quantification technique and classifier described in the previous section expresses the success of the underlying data treatment strategy. Logistic regressions return continuous probabilities of classification success hence one again must choose a threshold (usually 0.50) above which an outcome is considered successful in order to assess the regression's predictive ability against the actual data (i.e. binarized outcomes). Alternatively, one may evaluate classification success across the full range of probabilities in the same way done using ROC curves. Binary logistic regressions outcomes for each experimental replicate used to compute TSS scores were obtained by employing one round of 5-fold cross-validation per replicate. Because values above the binarization thereshold (i.e. the "ones") correspond to true positives and all other outcomes (true negatives, false positives and false negatives) were scored as "zeroes", each regression was evaluated using the area under the corresponding Precision-Recall curves (PRCs), and not ROCs, because the former quantifies the classifier positive predictive ability [34] (see graphical representation in S6-S9 Figs in S1 File). AUCs were estimated using the function *pr.curve* implemented in the R package *PRROC* [35].

## Statistical analyses

We used Generalized Linear Models (GLM) or Generalized Linear Mixed Models (GLMM) to test for differences across image treatment strategies. All analyses were conducted using the R package glmmTMB [36], model fit was assessed using Bayesian information criterion and also via residual analyses using DHARMA [37]. AUCs range bewteen 0 and 1, and TSS between -1 and 1, but all the scores of the latter statistic were on the positive side of the scale, thus, they were, for practical purposes, also bound betwwen 0 and 1. However, we treated both variables as normally- and not beta-distributed [38] because residual analysis revealed better fit to the data when we assumed the former distribution family. Whenever necessary, pairwise comparisons were conducted using Tukey's Honestly Significant Difference (HSD) *post-hoc* test performed on least-squares means estimates using the R package *emmeans* [39].

## Results

CLPB yielded 1,352 TD and 100 first Principal Components were retained for subsequent DAPC analysis. Classification score distributions obtained from the 5-fold cross-validation experiments for different image treatment strategies are summarized in Fig 2. The combination of CLBP and Θ-FAM yielded similar results both for *Porites* spp. and *Siderastrea* spp. in the sense that pertinence to the labeled species was, on average, higher than to alternative species (Fig 2a and 2b). This was not observed for CLBP+DAPC or when morphometric characters were submitted to Θ-FAM (Morph+Θ-FAM, Fig 2c and 2d). Probability distributions returned by DAPC were highly skewed towards human identified species, but had "fat tails" that spanned the full range of the distributions obtained for other species (Fig 2c). Skeweness was not as evident for Morph+Θ-FAM (Fig 2d) and medians of pertinence distributions were

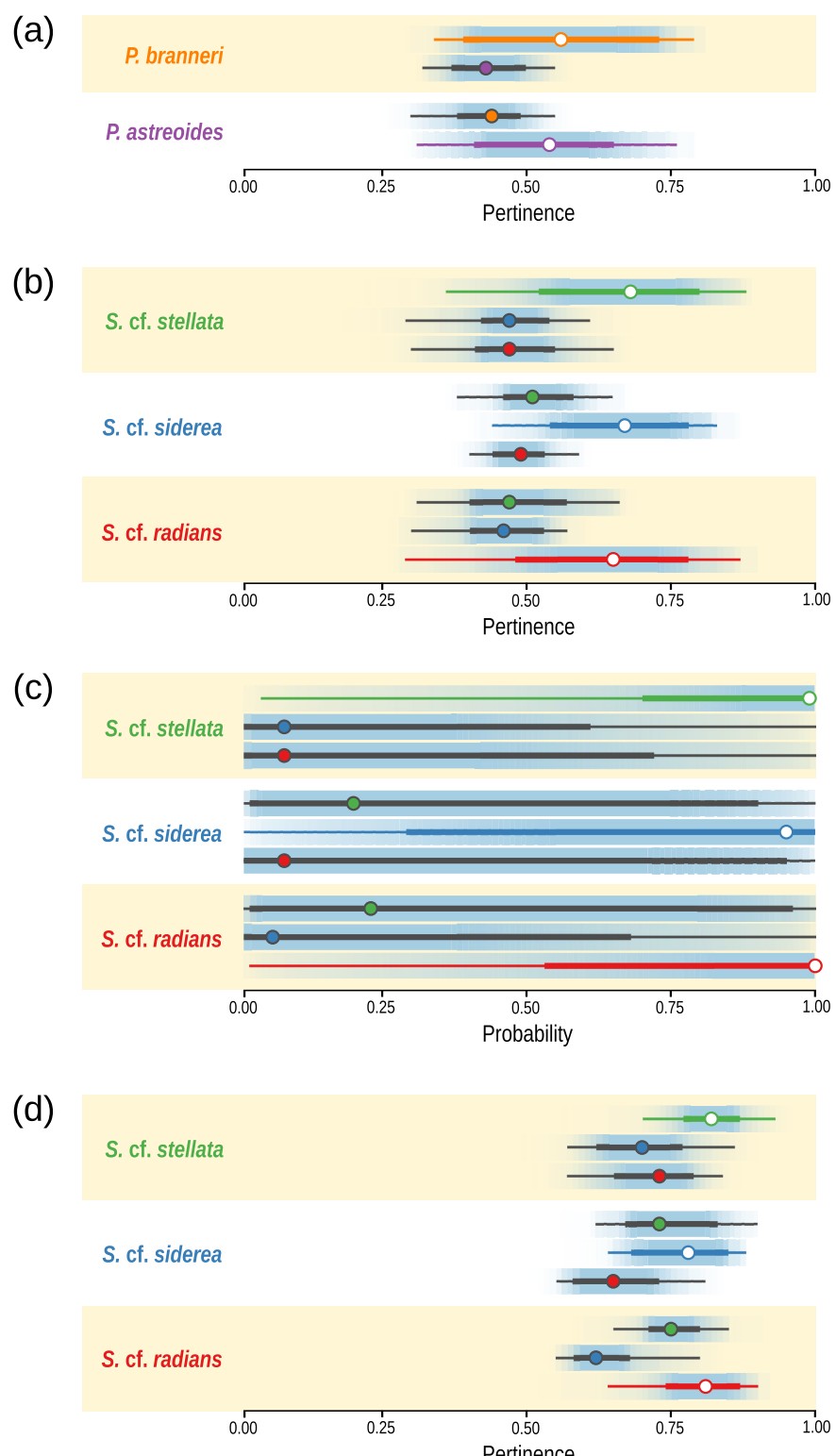

**Fig 2. Distributions of classification scores from 5-fold cross-validation experiments.** Circles represent median values, thicker bars are interquartile and thinner bars are non-outlier ranges. The darkest shade of the percentile ribbons correspond to 2nd and 3rd quartiles, becoming lighter towards the distribution tails. Scores for all species within each genus were generated for each image treatment strategy, grouped as separate subfigures. Range bars of distributions corresponding to correctly labeled outcomes are color-coded according to species name on the left side of

each panel. The other species are identified by the colored filling of the circles representing the median values of the remaining distributions within each panel. A. Pertinence scores obtained from Completed Local Binary Patterns (CLBP) sampled from *Porites* spp. images and classified using Θ-FAM; B. CLBP from *Siderastrea* spp. + Θ-FAM; C. Probability scores obtained from CLBP sampled from *Siderastrea* spp. images and classified using (linear) Discriminant Analysis of Principal Components (DAPC); D. Traditional morphometric descriptors sampled from *Siderastrea* spp. + Θ-FAM.

higher than those in Fig 2a and 2b, but once again distributions exhibited evident overlap, reflecting the low discriminatory power of those characters.

Classification performance results are summarized in Fig 3a. The distribution of TSS scores suggest that the performance of Θ-FAM was significantly worse when applied to CLBP sampled from *Porites* spp. images (the benchmark dataset) than it was for *Siderastrea* spp. (two leftmost panels on Fig 3a). We tested this statement by fitting Generalized Linear Mixed Models (GLMM) to the data, treating genera as levels in the fixed factor *Genus* and species as random factors nested within genera. Model comparison using Bayesian Information Criterion (BIC) favored the inclusion of a dispersion parameter to accommodate variance heterogeneity among *Genus* levels. The fixed effect estimate (slope) was highly significant (*estimate* = 0.16, *CI* = 0.12–0.20, $p < 0.001$, $n = 500$) and the residual analyses showed no deviance from the expected for the fitted model (S10 Fig in S1 File).

To test for differences in TSS scores among combinations of classifiers and descriptors applied to *Siderastrea* spp. only (three rightmost panels in Fig 3a), we fit Generalized Linear Models (GLM) to the data. Each combination of classifier and descriptor (i.e. CLBP+Θ-FAM, CLBP+DAPC and Morph+Θ-FAM), represented as separate panels in Fig 3a), was treated as a level of the fixed factor *Image Treatment* and *Siderastrea* species were the levels of the fixed factor *Species*. The best model according to BIC included an interaction term (i.e. *Image Treatment* x *Species*) and dispersion parameters for both fixed factors. Residual analyses using DHARMa supported model adequacy (S11 Fig in S1 File). All interaction terms were highly significant (see S2 Table in S1 File), so we used *post-hoc* Tukey's HSD to test for differences among all level combinations across fixed factors. **Non-significant** differences are marked by the same letters in Fig 3a and numeric results are reported in S3 Table in S1 File. DAPC performance was worse than Θ-FAM when both classifiers were applied to CLBP sampled from all species but *S.* cf. *stellata*, in which case classifier performances were indistinguishable (middle panels in Fig 3a). Image classification based on traditional morphometric descriptors was statistically worse than CLBP for all species-wise comparisons (Fig 3a).

Fig 3b shows variation among binarization thresholds (see Classifier performance—TSS for details) across image treatments. Variation in CLBP+DAPC among cross-validation replicates is striking when compared to thresholds distribution obtained using Θ-FAM, indicating overfitting, the underlying cause of the long tails in Fig 2c. The higher values of binarization thresholds in the treatment Morph+Θ-FAM reflects the higher pertinence scores observed in Fig 2d.

Binarized outcomes are represented in the lower panels of S6-S9 Figs in S1 File and binary logistic regressions expressing outcome probability as a function of classification scores (or how upper and lower panels in S6-S9 Figs in S1 File are correlated) are in S12 Fig in S1 File. Fig 2c shows the distributions of areas under Precision-Recall curves (PRCs) computed from binary logistic regressions trained on data from each replicate of the 5-fold cross-validation experiment. To statistically assess performance differences across image treatment strategies, we employed GLM. BIC once more supported the modelling of overdispersion and residual analyses using DHARMa showed no deviation from the expected under the chosen model (S13 Fig in S1 File). All predictors and contrasts among image treatments were highly

(a) (b) (c)

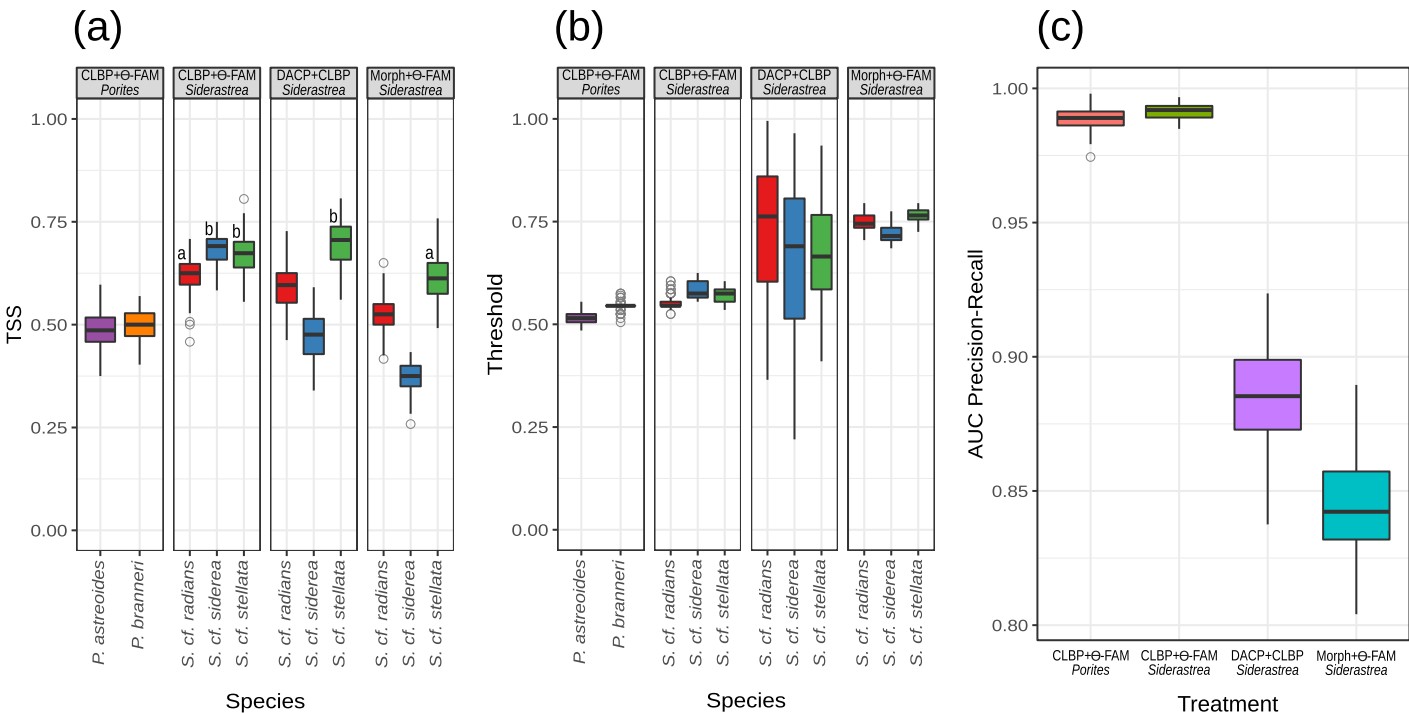

**Fig 3. Classifier performance metrics.** Horizontal bars represent medians, boxes are interquartile, vertical bars are non-outlier ranges and outliers are represented as circles. A. True Skill Statistic (TSS) computed for 100 replicates of the 5-fold cross validation experiments. Panels correspond to combinations of the two different image quantification strategies i.e. Completed Local Binary Patterns (CLBP) or traditional morphometrics, sampled from colonies of species belonging to either genus (i.e. *Porites* spp. or *Siderastrea* spp.) and classified using either Θ-FAM or (linear) Discriminant Analysis of Principal Components (DAPC). Boxes are color-coded according to species identified in the x-axis. Letters on the three rightmost panels indicate species distributions that are **not** significantly different from each other according to Tukey's HSD pairwise post-hoc test on least-square means. B. binarization thresholds corresponding to either pertinence (Θ-FAM) or probability values (DAPC) that maximize TSS scores represented in A. C. areas under Precision-Recall curves, computed after binarization of the cross-validation replicates using the thresholds in B. See Methods for additional details.

significant (S4 and S5 Tables in S1 File). Performance evaluation via meta-classifier was qualitatively different from the scenario obtained using TSS: CLBP+Θ-FAM outperformed other strategies when applied to *Porites* spp. and *Siderastrea* spp. Fig 2d.

## Discussion

We provided solid statistical evidence that the combination of computer vision and artificial intelligence outperformed traditional characters and classification techniques when it comes to species discrimination in the problematic genus *Siderastrea*. The employed dataset is certainly restricted when compared to the thousands of images normally used to train neural networks. This is likely reflected by the partial overlap among pertinence score distributions computed for the human identified and alternative species, particularly in the case of our benchmark (Fig 2a), but also in our working dataset (Fig 2b). Nevertheless, limited training was enough to ensure the superior performance of the new methods when compared to more commonly used approaches. CLBP yielded a much larger number of descriptors (n = 1352) than morphometric descriptors (n = 15) but, when subjected to DAPC, even the rich CLBP dataset was not enough to ensure good discrimination among the tested species (Fig 3a).

It is important to draw the distinction between employing LDA (or any other type of discriminant analysis) as an exploratory tool as opposed to a formal classifier, whose performance

must be assessed through cross-validation experiments. When applied in the first sense to *Siderastrea* CLPB, results are visually satisfactory, showing good discrimination among images in the morphospace defined by the two discriminant functions computed using DAPC (S14 Fig in S1 File). We could draw ellipses or polygons around each cluster and/or try to lend more credibility to these results by employing additional statistics such as Multivariate Analysis of Variance (MANOVA) [10]. The latter approach is non-advisable because may lead to the claim that classification is adequate simply because the probability of type I error is smaller than significance values (i.e. the proverbial p < 0.05). However, because power is related to sample size (and it is quite easy to obtain a very large number of measurements when working with corallites or images sampled from the same colony), significance may be obtained even when there is considerable overlap among species clusters. In other words, given a sufficiently large sample size, significance will almost always be achieved unless there is no species effect whatsoever on morphometric traits [40]. In the context of morphometric analysis, this means that samples belonging to different groups are indistinguishable from each other. Additionally, samples will likely violate the assumption of data independence because within-colony correlations among quantitative characters are expected to be higher than between-colony correlations; characters sampled from individuals belonging to the same morphotypes will be more correlated with each other than with those collected from other morphotypes, and so on [41]. Therefore, parametric tests are a poor measure of the classifier's true performance. Ignoring within-colony correlations means overlooking the modular nature of colonial organisms [42]. Still, a non-exhaustive survey of 23 articles on colonial cnidarians published between 1990 and 2018 shows that more than a third of the analyses (n = 7) used either MANOVA or PERMANOVA to validate their results (S6 Table in S1 File). While PERMANOVA may accommodate nested data structures [43], none of the studies reported implementation of hierarchical effects into their designs, and significance in the face of negligible effects when using this test is still a risk when working with large sample sizes.

Only 3 of those studies reported classification success, which is expected to be inflated if no cross-validation is employed because *a priori* probability of correctly assigning a specimen to 1 of $k$ species is $1/k$ [44, 45]. Three other studies did employ cross-validation (S6 Table in S1 File), but even those may have also produced inaccurate estimation of classifier's success. For instance, if 85% of the sampled **corallites** (or images, in our case) were successfully assigned to the expected cluster, it does not mean that 85% of the sampled **colonies** were correctly classified. This would be true only in the best case scenario in which corallite classification success is uniformly distributed across colonies, which is very unlikely (e.g. S6-S9 Figs in S1 File). Employing within-colony character averages [46, 47] instead of the raw data is not recommended because the researcher is throwing intra-colonial variance away [41], arithmetic means are influenced by outliers and are also a poor measure of central tendency when the character distributions are skewed [48].

None of the surveyed publications employed ML statistics based on confusion matrices, such as ROC curves or TSS, to evaluate classifier performance. This may also lead to inflated estimates of classification success, even when cross-validation is employed, if an outcome is considered successful simply because the highest probability corresponds to the expected class. For instance, if the posterior probabilities of cluster membership returned by a method such as LDA are 0.34, 0.33 and 0.33, this actually means that the classifier is rather unsure as where to place that particular sample. On the other hand, if results were 0.80, 0.15 and 0.05, the odds in favor of placement in the first cluster are 4 to 1. If the highest probabilities in both situations (i.e. 0.34 and 0.80) correspond to the correct cluster, and data are naively binarized using the aforementioned criterion, these outcomes will be rendered identical even though they correspond to very different probabilistic statements. One could argue that if the classifier is

confused, highest probabilities will be randomly distributed among all possible classes hence overall classification success will be low. This was not the case with Θ-FAM outcomes. S6 and S7 Figs in S1 File show that all images belonging to some of the sampled colonies (blocked in yellow in the lower panel) fell under the binarization threshold in every cross-validation experiment, although highest raw pertinence values correspond to the expected outcome, as evidenced by the color-coded scores in the upper panels. Although this opens up the interesting possibility that these colonies are morphological outliers, one cannot rule out the possibility of an artifactual result due to insufficient network training. At any rate, our brief literature encourages more rigorous evaluation of classification success in the morphometric analysis of Anthozoa.

Our results also uncovered surprising instability in DACP performance when applied to CLBP. The large variances of threshold distributions observed for all *Siderastrea* species (Fig 3b) means that the cutoff probability that maximizes the sum of DACP's specificity and sensitivity varied considerably among replicates in cross-validation experiments, due to overfitting of discriminant functions to the training sets. This is a rather undesirable property of machine learning algorithms because it implies in poor generalization ability i.e., extrapolation beyond the domain of values used to train the classifier [45]. Because probability cutoffs were used to binarize the data, the incidence of positive outcomes as a function of posterior probabilities will also vary across experimental replicates. This is reflected in the meta-learning performance evaluation by the lower precision-recall AUCs for DAPC when compared to those obtained for the Θ-FAM (Fig 3c. AUCs were obtained using one additional round of 5-fold cross-validation trials on each previously computed experimental replicate. In each of the 5 rounds, binary logistic regressions expressing classification success as a function of classification scores were trained on 80% of the data and used to predict successful outcomes in the remaining 20% of the data making up the validation set (see Classifier performance—Meta-learning for details). Since discriminant functions were overfitted to the training sets in the original cross-validation experiments, outcome success as a function of posterior probabilities will also be highly variable across those replicates. Put another way, when classification scores are good predictors of classification outcomes, one should be able to "draw a line" across score distributions for all species and find that values equal or higher than this threshold are only found for correctly labeled images. It is easy to see how it is possible to draw such lines across Fig 2a and 2b, but not across Fig 2c, because distributions overlap so much. Red lines if S12 Fig in S1 File correspond to the theoretical threshold above which 99% of the outcomes are correctly labeled (i.e. true positives). This line is not present in (c) because the threshold exceeds 1, which is the maximum theoretical value for probabilities.

Overall, the worst image treatment approach to species identification in *Siderastrea*, according to both classification performance strategies, was the combination of traditional morphometric descriptors and Θ-FAM (Morph.+Θ-FAM, see Fig 3a and 3c). Unlike DAPC, the lower performance of Morph.+Θ-FAM is not a result of overfitting, as evidenced by the tight distributions of the corresponding binarization thresholds (Fig 3b). It is most likely due to the overlap of those characters across species, which confuses the classifier as much as it would confuse a novice taxonomist that relies solely on distributions and counts when attempting to identify specimens. This uncertainty is expressed by the superposed pertinence distributions in Fig 2d, that also have higher medians than those obtained when Θ-FAM was applied over CLBP (Fig 2b and 2c). Higher medians are also a function of the uncertain boundaries among character ranges, as illustrated by the following (contrived) example. Suppose that a classification algorithm had to distinguish among cats, bats and humans based on features such as number of eyes, ears, nostrils, limbs, digits, and the presence or absence of mammary, sweat and sebaceous glands. In cladistic parlance, these traits are simplesiomorphies. Therefore, because all

those animals share the presence of the aforementioned glands and have two eyes, two ears, two nostrils, four limbs and five digits, any specimen presented to the classifier would have high pertinence to either species and, precisely because of that, it would be impossible to distinguish among them using that short list of characters. As a consequence of distribution superposing, the aforementioned pertinence threshold should have a value close to 1 (Fig 2d), as illustrated by the red line in S12 Fig in S1 File (d). We propose that the contrast between Θ-FAM's performance when applied to CLBP versus its performance when applied to morphometric characters mirrors the human ability to visually discriminate among species using skeletal texture versus human inability to correctly express these differences trough ranges and/or frequencies of continuous and/or discrete characters.

As demonstrated by the TSS scores (Fig 3a), Θ-FAM performed worst when applied to *Porites* spp. images, although human species identification was not an issue in this benchmark dataset. These results are in contrast with the performance measured used the meta-learning approach (Fig 3c), that again suggests no overfitting. The benchmark dataset had the fewest images and *Porites* spp. specimens also displayed significant morphological plasticity, as exemplified in S2 Fig in S1 File), that shows similar specimens belonging to different species (a-d) and, conversely, specimens belonging to the same species and collected from similar environments with radically different morphologies (e-h). Extreme morphological plasticity of corallite architecture in Atlantic *Porites* spp. colonies, and trouble employing these characters to resolve taxonomy in the genus have long been recognized [49], but subsequent studies, although subjected to the aforementioned caveats, have reportedly been more successful [46, 50]. Texture-based classification in *Porites* spp. is certainly challenging, so this seems to be an appropriate test genus for further research into supervised species discrimination in Scleractinia.

## Conclusions

In this study we have, for the first time, applied computer vision and fuzzy neural networks to the problem of species discrimination in hard corals. Our results suggest that this approach is effective in emulating human discernment, thus bringing the task closer to the taxonomist's perspective: when presented with morphometric characters traditionally used in taxonomy, Θ-FAM returns high values of pertinence to the either species, reproducing the confusion of the unskilled end user working with a few quantitative characters with overlapping ranges. On the other hand, when trained with descriptors that attempt to emulate the visual impression of experienced taxonomists, Θ-FAM's discrimination ability was significantly improved. We have also introduced a novel method to evaluate classifier performance based on meta-learning that uncovered surprising instability in a commonly used multivariate technique. While one could rightfully argue that there are better choices of diagnostic characters than those sampled in this study, computer vision allows for the collection of large volumes of information in a much shorter period of time than labor-intensive manual measurements. Additionally, it eliminates the problem of subjectivity in character choice and/or the issues of comparability when these characters are to be collected across morphologically divergent species. Coral taxonomists have more recently added microstructural and micromorphological features to the suite of diagnostic characters used in the description of new species and/or taxonomic revisions (e.g. [51]). These latter features are also ordinarily sampled via SEM hence should readily lend themselves to quantification using texture descriptors such as CLBP. We also collected images at two different spatial scales (corallite and colony) and integrated them into a single analysis, suggesting that images taken at a variety of magnifications, describing microstructure and micro/macromorphology, could also be analyzed together.

The approach described in this study was geared towards species recognition and requires supervision in the sense that species must have been previously described. Therefore, new species description is still a human job and it is not our intention to suggest that IA should or will replace taxonomists. Nevertheless, recent advances in DNA sequencing technology have revealed that cryptic speciation is rampant across the Tree of Life [52], particularly in marine environments [53]. Crypitcness, however, lies in the eye of the beholder. Classifiers may be trained using TD sampled from specimens labeled as genetically delimited species and subsequently tested in order to see how effective they are in recognizing these boundaries, revealing how cryptic speciation really is. Alternatively, one could use unsupervised classifiers to morphologically cluster specimens and compare the resulting pattern to the corresponding molecular phylogeny/phylogeography. This approach has been commonly applied to Scleractinia using multivariate techniques as classifiers (e.g [54]) albeit with the aforementioned statistical caveats and considerable time spent on manual data collection.

Fuzzy logic was employed to accommodate our own uncertainty about species assignment, but texture descriptors may, of course, be used independently of fuzzy ANNs, because several other machine learning algorithms could be employed as classifiers. Conversely, E-FAMs may be used to discriminate among operational taxonomic units (OTUs) or other entities using different descriptive data: in our implementation (See Data Availability), CLBP generation and $\Theta$-FAM classification are conducted independently of each other. E-FAMs should be considered as candidate classifiers in those situations where human drawn boundaries are not as hard as we would like them to be. For instance, one may attempt to characterize environmentally induced variation in colonial morphology across a fringing reef by collecting (and labelling) specimens from the reef flat, reef crest and fore reef. If specimens are collected along a transect, transitions between habitat may not be as sharp as the sampling design may convey. The same would be true for geographically distinct populations that maintain some level of genetic connectivity (i.e. metapopulations), specimens collected along environmental (e.g. depth) gradients, know hybridization zones, etc. Fuzzy logic was invented to deal with fuzzy boundaries, which are much more prevalent in nature than is normally acknowledged.

## Supporting information

**S1 File.**
(PDF)

## Acknowledgments

We thank Dr. André Guaraldo for the help with GLMM and two anonymous reviewers for valuable comments on earlier drafts.

## Author Contributions

**Conceptualization:** Marcos Soares Barbeitos, João Batista Florindo, Estevão Esmi Laureano.

**Data curation:** Marcos Soares Barbeitos, Flávio Alberto Pérez, Julián Olaya-Restrepo, Ana Paula Martins Winter.

**Formal analysis:** Marcos Soares Barbeitos, Flávio Alberto Pérez, João Batista Florindo, Estevão Esmi Laureano.

**Funding acquisition:** Marcos Soares Barbeitos, João Batista Florindo, Estevão Esmi Laureano.

**Investigation:** Marcos Soares Barbeitos.

**Methodology:** Marcos Soares Barbeitos, Flávio Alberto Pérez, Julián Olaya-Restrepo, Ana Paula Martins Winter, João Batista Florindo, Estevão Esmi Laureano.

**Project administration:** Marcos Soares Barbeitos.

**Resources:** Marcos Soares Barbeitos, Julián Olaya-Restrepo, Ana Paula Martins Winter.

**Software:** Marcos Soares Barbeitos, João Batista Florindo, Estevão Esmi Laureano.

**Supervision:** Marcos Soares Barbeitos.

**Validation:** Marcos Soares Barbeitos, João Batista Florindo, Estevão Esmi Laureano.

**Visualization:** Marcos Soares Barbeitos.

**Writing – original draft:** Marcos Soares Barbeitos, Flávio Alberto Pérez, Ana Paula Martins Winter.

**Writing – review & editing:** Marcos Soares Barbeitos, João Batista Florindo, Estevão Esmi Laureano.

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
