## [Decision Letter · Decision Letter 0]

6 Feb 2024

PONE-D-23-11242AI based coral species discrimination: a case study of the Siderastrea Atlantic ComplexPLOS ONE

Dear Dr. Barbeitos,

Thank you for submitting your manuscript to PLOS ONE. After careful consideration, we feel that it has merit but does not fully meet PLOS ONE’s publication criteria as it currently stands. Therefore, we invite you to submit a revised version of the manuscript that addresses the points raised during the review process.

We look forward to receiving your revised manuscript.

Kind regards,

Clara Sousa

Academic Editor

PLOS ONE

Journal Requirements:

3.  Please note that your Data Availability Statement is currently missing the repository name and/or the DOI/accession number of each dataset OR a direct link to access each database. If your manuscript is accepted for publication, you will be asked to provide these details on a very short timeline. We therefore suggest that you provide this information now, though we will not hold up the peer review process if you are unable.

6. We note that Figure 1 in your submission contain map/satellite images which may be copyrighted. All PLOS content is published under the Creative Commons Attribution License (CC BY 4.0), which means that the manuscript, images, and Supporting Information files will be freely available online, and any third party is permitted to access, download, copy, distribute, and use these materials in any way, even commercially, with proper attribution. For these reasons, we cannot publish previously copyrighted maps or satellite images created using proprietary data, such as Google software (Google Maps, Street View, and Earth). For more information, see our copyright guidelines: http://journals.plos.org/plosone/s/licenses-and-copyright.

Reviewers' comments:

Reviewer's Responses to Questions

**Comments to the Author**

1. Is the manuscript technically sound, and do the data support the conclusions?

Reviewer #1: Yes

Reviewer #2: Yes

2. Has the statistical analysis been performed appropriately and rigorously? 

Reviewer #1: Yes

Reviewer #2: Yes

3. Have the authors made all data underlying the findings in their manuscript fully available?

Reviewer #1: No

Reviewer #2: Yes

4. Is the manuscript presented in an intelligible fashion and written in standard English?

Reviewer #1: Yes

Reviewer #2: Yes

5. Review Comments to the Author

Reviewer #1: The manuscript uses automated quantification and machine learning to identify species, focusing on the genus Siderastrea. The proposed method is interesting, but there are a few issues that I believe that should be addressed. Please find below some general comments and suggestions.

The authors started the manuscript by raising a relevant discussion about the challenges of species identification. However, in the way that the first paragraph is presented, it seems that taxonomic descriptions are unreliable. I can see the utility of applying new methods to facilitate and improve species delimitation, but the taxonomic work is not the source of the problem. Furthermore, the term “taxonomic impediment” means a lot more than the difficult faced by end users (which it is not listed within the “Real taxonomic impediments” in the cited work of Ebach et al. 2011) and I don’t think the term is properly address by the authors (it is at least oversimplified). Citing Ebach et al. (2011) “It is erroneous to think that taxonomy can be replaced by taxonomic tools or technological surrogates. Such issues are merely applications for taxonomic information.”

I understand the concern raised by Reviewer #2 regarding not knowing the true species identity. Maybe if the analysis were also performed in the type species this issue could be better addressed in the manuscript and it would shed some light on the species delimitation within Siderastrea.

How to deal with geographic differences? The measurements provide in Figure S1 show, in some cases, a strong divergence between samples from Brazil and Gulf of Mexico. Some species have similar morphology due to environmental constrains and are difficult to separate based solely on morphological characters. I wonder how the proposed analysis would handle these issues.

Furthermore, an important question that could be more clearly addressed about the utility of the tools (a concern raised by reviewer #1) is if the method helped to clarify undistinguishable specimens.

Why not include SEM images for Porites but not the main target species? Example images used during the study would also be interesting.

Why the same species used for CLBP were not used for the morphological analysis? This would make the method more comparable. Furthermore, a table with all morphological measurements should be provided.

Line 4: Define SI

Lines 27-30: I can see the overlap between S. siderea from Panama and S. stellata from Paraíba on the distance among columella and the number of septa, but for the corallite diameter there is an overlap between S. siderea from Panama and S. stellata from Gulf of Mexico.

Line 129: Did you mean S9 Fig?

Line 140: Why did you chose to use the word “sclerits” here? I believe that corallites might be a better fit here.

Lines 141-144: Does this mean that you didn’t evaluate the morphology of these specimens? This does not make sense.

Line 177: I’m not sure what do mean with “sclerite diameter”

Line 185: Do you mean Table S5? Why are the measures not provided?

Line 367: Do you mean S11?

Table 1: Include the full definition of CLBP.

Supplementary Information

Line 19: I believe the correct citation is from Milne-Edwaurds (1857).

Reviewer #2: Dear authors:

This study presents significant advancements in the field of species delimitation in hard corals, employing computer vision and fuzzy neural networks. One of the primary contributions is the demonstrated efficacy in species discrimination, nearing human perception and offering a potential solution to the taxonomic challenges faced in this group. Moreover, the ability to collect large volumes of data more efficiently through computer vision promises to expedite the process of taxonomic identification and revisions. The integration of images at different spatial scales also represents a breakthrough, allowing for a more comprehensive analysis of microstructural and micromorphological features. However, the study reveals limitations in traditional approaches due to the inability to distinguish some Siderastrea species, indicating the need for improvement in identification techniques. Additionally, the reliance on previous human descriptions for species identification and the inherent uncertainties in classification underscore the ongoing importance of human involvement in this process. The findings are still preliminary due to limited algorithm training, highlighting the need for further research and refinement to achieve robust and reliable results.

Positive Points:

Efficiency in Species Discrimination: The application of computer vision and fuzzy neural networks in this study showed promising results in emulating human perception, indicating effectiveness in species discrimination. This suggests a potential breakthrough in overcoming taxonomic challenges in hard corals.

Enhanced Data Collection: Computer vision enables the collection of large volumes of information in a much shorter period compared to labor-intensive manual measurements. This advancement could significantly expedite the process of species identification and taxonomic revisions.

Integration of Multiple Spatial Scales: The integration of images taken at different spatial scales (corallite and colony) into a single analysis suggests a comprehensive approach to studying microstructural and micromorphological features, potentially improving accuracy in species identification.

Potential for Larger Scale Application: The abundance of labeled SEMs (scanning electron micrographs) available in published literature indicates the potential for larger-scale application of automated species recognition, which could facilitate efforts similar to freely available platforms like iNaturalist, eBird, and BirdNet.

Revealing Cryptic Speciation: The use of artificial neural networks (ANNs) and other methods trained on morphological traits sampled from genetically delimited species could reveal the extent of cryptic speciation, shedding light on previously unrecognized species boundaries.

Negative Points:

Limitations of Traditional Methods: Conventional morphometric data and sophisticated artificial intelligence algorithms have shown limitations in discriminating among Siderastrea species, indicating the need for improved approaches in species identification.

Dependency on Human Descriptions: While the combined approach described in the study shows promise for species recognition, it still relies on previously described species, highlighting the continued necessity for human involvement in new species description.

Uncertainties in Classification: Despite employing fuzzy logic to accommodate uncertainty about species assignment, there are limitations and potential biases inherent in the classification process, which could affect the accuracy of results.

Statistical Caveats: Previous approaches using multivariate techniques for species classification in Scleractinia have been subject to statistical caveats and considerable time spent on manual data collection, indicating challenges in achieving robust and reliable results.

Limited Training Data: The tentative nature of the results due to limited network training suggests that further research and refinement are necessary to fully realize the potential of automated species recognition in hard corals.

Materials and methods

I suggest that individual voucher numbers be included for each collected specimen in order to provide a unique and traceable identification for each sample. This will aid in organizing the data associated with each specimen and facilitate future referencing and research. Please consider inserting these voucher numbers into the document for a more comprehensive and detailed documentation of the collected samples.

A flaw here is the lack of clarity in the identification of coral species, especially those of the Siderastrea genus. Although the text mentions that the species were identified according to the most recent taxonomic treatment of the genus, it also indicates that the identification was done tentatively due to the lack of clear diagnostic features. This means that there may be uncertainty about the accuracy of Siderastrea species identifications. Additionally, the presence of specific diagnostic features for some species (such as the presence of intra-tentacular budding for S. stellata) may not be conclusive, which can lead to errors in species identification. Therefore, the flaw lies in the lack of definitive methods or diagnostic characteristics to accurately identify coral species.

6. PLOS authors have the option to publish the peer review history of their article (what does this mean?). If published, this will include your full peer review and any attached files.

Reviewer #1: No

Reviewer #2: No

---

## [Author Response · Author response to Decision Letter 0]

3 Jun 2024

Journal Requirements

1. “Please ensure that your manuscript meets PLOS ONE's style requirements, including those for file naming. The PLOS ONE style templates can be found at...”

R. All references to figures in text were replaced by “Fig” instead of “Fig.” and files were renamed according to PLoS standards. SI figures were also renumbered according to their order of appearance in the main text.

2. “Please note that funding information should not appear in any section or other areas of your manuscript. We will only publish funding information present in the Funding Statement section of the online submission form. Please remove any funding-related text from the manuscript.”

R. Funding information was removed from Acknowledgements.

3. “Please note that your Data Availability Statement is currently missing the repository name and/or the DOI/accession number of each dataset OR a direct link to access each database. If your manuscript is accepted for publication, you will be asked to provide these details on a very short timeline. We therefore suggest that you provide this information now, though we will not hold up the peer review process if you are unable.”

R. Images, CSV files with morphometric measurements and textute descriptors were all deposited in Morphobank and will be freely available after paper acceptance. Matlab/Octave code with detailed instructions will also be freely available from Github after paper acceptance.

5. “We note that you have included the phrase “data not shown” in your manuscript. Unfortunately, this does not meet our data sharing requirements. PLOS does not permit references to inaccessible data. We require that authors provide all relevant data within the paper, Supporting Information files, or in an acceptable, public repository. Please add a citation to support this phrase or upload the data that corresponds with these findings to a stable repository (such as Figshare or Dryad) and provide and URLs, DOIs, or accession numbers that may be used to access these data. Or, if the data are not a core part of the research being presented in your study, we ask that you remove the phrase that refers to these data.”

R. We have added the required information to the text.

6. We note that Figure 1 in your submission contain map/satellite images which may be copyrighted. All PLOS content is published under the Creative Commons Attribution License (CC BY 4.0), which means that the manuscript, images, and Supporting Information files will be freely available online, and any third party is permitted to access, download, copy, distribute, and use these materials in any way, even commercially, with proper attribution. For these reasons, we cannot publish previously copyrighted maps or satellite images created using proprietary data, such as Google software (Google Maps, Street View, and Earth). For more information, see our copyright guidelines: http://journals.plos.org/plosone/s/licenses-and-copyright.

R. We have replaced the map using imagery from NASA Earth Observatory (public domain http://earthobservatory.nasa.)

Reviewer #1 

1. “The authors started the manuscript by raising a relevant discussion about the challenges of species identification. However, in the way that the first paragraph is presented, it seems that taxonomic descriptions are unreliable. I can see the utility of applying new methods to facilitate and improve species delimitation, but the taxonomic work is not the source of the problem. Furthermore, the term “taxonomic impediment” means a lot more than the difficult faced by end users (which it is not listed within the “Real taxonomic impediments” in the cited work of Ebach et al. 2011) and I don’t think the term is properly address by the authors (it is at least oversimplified). Citing Ebach et al. (2011) “It is erroneous to think that taxonomy can be replaced by taxonomic tools or technological surrogates. Such issues are merely applications for taxonomic information.”

R. We agree with the reviewer. We cited Ebach et al. 2011 in the context of proper training of end users as a major source of taxonomic impediment. We attempted to clarify this by changing the sentence in Introduction from:

“To be fair, this is certainly a feature of many taxonomic keys and the trouble non-specialists face going through them has been termed ‘taxonomic impediment to end users’”

to:

“To be fair, this is certainly a feature of many taxonomic keys and the trouble that those who lack proper taxonomic training face going through them has been termed ‘taxonomic impediment to end users’”

We also added verbiage to the first paragraph of the Conclusion to stress that the approach used in this paper is not intended to replace taxonomic work and removed the last paragraph to eliminate any suggestion that taxonomic impediment will be eliminated by technology and not by properly trained taxonomists.

2. “I understand the concern raised by Reviewer #2 regarding not knowing the true species identity. Maybe if the analysis were also performed in the type species this issue could be better addressed in the manuscript and it would shed some light on the species delimitation within Siderastrea.”

R. We agree with both Reviewers. Unfortunately, there are three major issues here. The first is that, among the Atlantic species, only Siderastrea glynni (a synonym of S. siderea) has a designated (holo) type specimen. Second, the method requires removing a sample from the corallum hence it could be destructive for smaller type specimens. Third, the evidence presented in S1 Fig suggests that there is considerable geographic variation in diagnostic characters, thus a single holotype per species would not adequately capture the range of morphological variation present in species of this genus.

3. “How to deal with geographic differences? The measurements provide in Figure S1 show, in some cases, a strong divergence between samples from Brazil and Gulf of Mexico. Some species have similar morphology due to environmental constrains and are difficult to separate based solely on morphological characters. I wonder how the proposed analysis would handle these issues.”

R. That is precisely the role of fuzzy logic. This approach acknowledges that hard boundaries among human designated categories (species, in our case) may actually not exist and incorporates this uncertainty into the discrimination process. If geographic and/or environmentally introduced variation is such that hard bounds cannot really be drawn, pertinence scores will also reflect that as This was clearly demonstrated in the case of Porites (see Fig 2 and S6 Fig): even though there is no doubt about species assignment in this genus due to the obvious differences in color and colony shape, discrimination would be much harder if one was to rely solely on corallite morphology, as evidenced in S2 Fig. 

4. “Furthermore, an important question that could be more clearly addressed about the utility of the tools (a concern raised by reviewer #1) is if the method helped to clarify undistinguishable specimens.”

R. We believe to have clearly addressed this concern in the paper by showing that the combination of CLBP and Θ-FAM outperformed the alternative classifier most commonly applied to such problems: the instability of DAPC (an improved variant of LDA), uncovered by rigorous application of ML performance evaluation, shows that the seemingly satisfactory (and certainly overrated) results in S14 Fig are a consequence of inadequate application of the method. We invite the reviewers to appreciate the fact that we could have very well identified specimens to the best of our knowledge and claimed success after those results, as our brief survey suggests that most researchers would do. Instead, we attempted to incorporate the fuzziness in morphological boundaries introduced by geographic/environmental variation, compounded with our very uncertainty about specimen identification into the discrimination process, not as an attempt to devise a foolproof method of species discrimination, but rather to prevent ourselves from being overconfident in our own results.

As to the choice of this particular set of species, we have stated in the Introduction that our approach was problem-oriented in the sense that, as end-users, we were having a hard time identifying our specimens with the best taxonomic treatment at hand. The evidence in S1 Fig actually highlights the need for a comprehensive revision of the genus Siderastrea and the designation of whole syntypic series in order to adequately capture morphological variation across the Atlantic Ocean. Therefore, uncertainty in species identification should challenge even experienced taxonomists and yet, some consistent variation does exist, otherwise results would be statistically similar to or worse than those obtained for Porites (see Fig. 2). At any rate, the success of our approach with respect to other species certainly deserves further testing by the community at large, thus we strongly believe this manuscript to be worthy of publication. 

We have also offered two additional contributions, which may be employed independently of each other: the first is the introduction of texture descriptors as means of quantifying SEMs in a fast, objective and reproducible way across a broad taxonomic range. The second is the use of E-FAMs as classifiers that allow researchers to deal with fuzzy boundaries among whatever classes they decide to bin their samples into. In this paper, fuzziness was a byproduct of the taxonomic complexity inherent to quantitative characters, but it could very well arise from biological realities, such as environmental gradients, hybridization zones, metapopulations, etc. 

Finally, we believe to have also succeeded in achieving the main goal of this paper by showing that fuzzy associative memories do emulate the human perception of species boundaries in the Siderastrea complex. When presented only with traditional characters, Θ-FAM gets “confused” (Fig. 2d) and finds specimens to have high pertinence to either species simply because these traits offer poor discrimination, making it hard do “draw lines” among them, as statistically illustrated by logistic regressions (see S12d Fig). When Θ-FAM was presented with CLBP, which approximates the visual experience of texture, “drawing the line” becomes possible because any image whose pertinence exceeded 0,70 was assigned to the “correct” species (see S12b Fig). We have thoroughly restructured the Conclusions in an attempt to make all these contributions more evident to prospective readers.

5. “Why not include SEM images for Porites but not the main target species? Example images used during the study would also be interesting.”

R. We added sample images, please see S4 Fig.

6. “Why the same species used for CLBP were not used for the morphological analysis? This would make the method more comparable. Furthermore, a table with all morphological measurements should be provided.”

R. We did not sample morphometric characters from Porites because they are notoriously uninformative and the inclusion of colonies of this genus in our sampling was intended to benchmark CLBP only. We could not acquire CLBP from all Siderastrea spp. individuals mainly due to our limited access to the microscopy facility (some important deadlines would not be met if we tried to SEM all specimens). The lack of CLBP sampling from all colonies actually makes our hypothesis testing more conservative: because ANN performance tends to increase with the amount of training, the fact that more images were analyzed for morphometric characters than they were for CLBP should theoretically benefit the performance of Θ-FAM in the former case.

7. “Line 4: Define SI”

R. Done

8. “Lines 27-30: I can see the overlap between S. siderea from Panama and S. stellata from Paraíba on the distance among columella and the number of septa, but for the corallite diameter there is an overlap between S. siderea from Panama and S. stellata from Gulf of Mexico.”

R. We thank the reviewer for pointing this out. The color of the bar was incorrect, Santos et al. did not sample any colonies from Panama. We have fixed the figure and now it reflects what was written in the Introduction. 

9. “Line 129: Did you mean S9 Fig?”

R. Fixed, now it is S2 Fig, thank you.

10. Line 140: Why did you chose to use the word “sclerits” here? I believe that corallites might be a better fit here.

R. We meant corallites, it was fixed, thank you for your careful reading.

11. “Lines 141-144: Does this mean that you didn’t evaluate the morphology of these specimens? This does not make sense.”

R. We did evaluate their morphology: we collected morphometric data from these specimens (Table 1) and they conformed well to the ranges reported for S. stellata in the literature (see S3 Fig and S5 Fig). According to the taxonomic status quo for the genus, only S. stellata has ever been described for Búzios. All specimens were collected from the same site (the beach spans only a few hundred feet) and have exceptionally large sizes, further increased by colony fusion, as remarked in the previous version of the manuscript. Búzios is subjected to upwelling, so the average temperature of sea water is lower than what is reported throughout the coast to the north of this city, which is the southernmost site in the range of Brazilian symbiotic hard corals (the only other Scleractinian species found at this site is the endemic Mussismilia hispida). These particularities make the occurrence of any other Siderastrea species at this site unlikely. Nevertheless, we do see how the statement at the end of the “Species identification” section may lead the reader astray, so we have removed it from this version.

12. “Line 177: I’m not sure what do mean with “sclerite diameter””

R. Once again, we meant corallite, thanks for pointing this out. 

13. “Line 185: Do you mean Table S5? Why are the measures not provided?”

R. We have carefully revised an reordered references to figures and tables in the SI, so we believe it has been fixed in this version. Images, morphometric measurements and CLBP descriptors were uploaded to Morphobank (www.morphobank.org).

14. “Line 367: Do you mean S11?”

R. See previous response, we thank the reviewer for pointing out this issue.

15. “Table 1: Include the full definition of CLBP.”

R. Done

16. “Supplementary Information Line 19: I believe the correct citation is from Milne-Edwaurds (1857).”

R. Fixed, thank you.

Reviewer #2

1. “I suggest that individual voucher numbers be included for each collected specimen in order to provide a unique and traceable identification for each sample. This will aid in organizing the data associated with each specimen and facilitate future referencing and research. Please consider inserting these voucher numbers into the document for a more comprehensive and detailed documentation of the collected samples.”

R. Done, please see Table S7.

2. “A flaw here is the lack of clarity in the identification of coral species, especially those of the Siderastrea genus. Although the text mentions that the species were identified according to the most recent taxonomic treatment of the genus, it also indicates that the identification was done tentatively due

---

## [Decision Letter · Decision Letter 1]

8 Oct 2024

AI based coral species discrimination: a case study of the Siderastrea Atlantic Complex

PONE-D-23-11242R1

Dear Dr. Barbeitos,

We’re pleased to inform you that your manuscript has been judged scientifically suitable for publication and will be formally accepted for publication once it meets all outstanding technical requirements.

Kind regards,

Clara Sousa

Academic Editor

PLOS ONE

Reviewers' comments:

Reviewer's Responses to Questions

**Comments to the Author**

1. If the authors have adequately addressed your comments raised in a previous round of review and you feel that this manuscript is now acceptable for publication, you may indicate that here to bypass the “Comments to the Author” section, enter your conflict of interest statement in the “Confidential to Editor” section, and submit your "Accept" recommendation.

Reviewer #2: All comments have been addressed

Reviewer #3: All comments have been addressed

2. Is the manuscript technically sound, and do the data support the conclusions?

Reviewer #2: Yes

Reviewer #3: Yes

3. Has the statistical analysis been performed appropriately and rigorously? 

Reviewer #2: Yes

Reviewer #3: Yes

4. Have the authors made all data underlying the findings in their manuscript fully available?

Reviewer #2: Yes

Reviewer #3: Yes

5. Is the manuscript presented in an intelligible fashion and written in standard English?

Reviewer #2: Yes

Reviewer #3: Yes

6. Review Comments to the Author

Reviewer #2: (No Response)

Reviewer #3: (No Response)

7. PLOS authors have the option to publish the peer review history of their article (what does this mean?). If published, this will include your full peer review and any attached files.

Reviewer #2: No

Reviewer #3: No

---

## [Editor Report · Acceptance letter]

18 Oct 2024

PONE-D-23-11242R1 

PLOS ONE

Dear Dr. Barbeitos, 

I'm pleased to inform you that your manuscript has been deemed suitable for publication in PLOS ONE. Congratulations! Your manuscript is now being handed over to our production team.

Kind regards, 

on behalf of

Dr. Clara Sousa 

Academic Editor

PLOS ONE